# Study of the Polishing Characteristics by Abrasive Flow Machining with a Rotating Device

Ken-Chuan Cheng [1], A-Cheng Wang [2],*, Kuan-Yu Chen [1] and Chien-Yao Huang [3]

1 Department of Mechanical Engineering, Chung-Yuan Christian University, Taoyuan 320314, Taiwan; kccheng20@gmail.com (K.-C.C.); gychen@cycu.edu.tw (K.-Y.C.)
2 Department of Mechanical Engineering, Chien-Hsin University of Science and Technology, Taoyuan 320312, Taiwan
3 Taiwan Instrument Research Institute, National Applied Research Laboratories, Hsinchu 300092, Taiwan; msyz@narlabs.org.tw
* Correspondence: acwang@uch.edu.tw

**Abstract:** Since only uni-direction motion is produced by traditional abrasive flow machining (AFM), so the polishing effects of the inner hole is not easy to achieve uniform roughness of the whole surface after polishing. Therefore, in this study, a rotating device with a DC servo motor was set up in the AFM to increase the tangential forces on the machining surface, and therefore, improve the uniform surface roughness and polishing efficiency. The rotating device was designed by a group of transmission gear set and a DC servo motor to create a rotational finishing path for the abrasive medium. The rotational motion of an abrasive can create different tangential forces on the working surface, inducing a more complex polishing path than that of traditional AFM. In addition to rotational speed, a servo motor can also change rotation directions in one working process, causing an abrasive medium to create many irregular finishing paths in the AFM. The experimental results showed that the surface roughness of the workpiece was significantly decreased with an increase in the rotational speed. Additionally, the results also showed that the surface roughness (SR) of the inner hole decreased from 0.61 μm Ra to 0.082 μm Ra after 20 machining cycles, the surface roughness improvement rate reached 87% at 15 rpm rotational speed, by applying a 1.5:1 silicone gel/abrasive concentration ratio and #60 abrasive mesh in the experiments. This study created excellent polishing efficiency by using a servo rotational device with AFM to produce good surface quality.

**Keywords:** abrasive flow machining; rotational device; polishing efficiency; surface roughness; radial force; axial force; tangential forces; servo motor



## 1. Introduction

The polishing characteristics of AFM include a simple finishing process for easily polishing irregular and complex surfaces and creating good quality final products. Jain et al. [1] conducted an investigation to identify the effectiveness of increasing material removal (MR) and decreasing surface roughness by modifying some key input parameters of the AFM process. Wang [2–5] developed semi-solid gels mixed with abrasives to apply as abrasive media in AFM, with two reciprocating hydraulic cylinders that could push the gel abrasives through passageways to finish complex surfaces. Since the inside of the gel was filled with a high number of abrasive particles, such as silicon carbides or alumina oxides, it became a high viscosity gel medium that could induce much more intensive shear forces to abrade the working surface, and therefore, the working surface could be polished effectively during the reciprocating motion in AFM. AFM also has the ability to remove machining burrs as well as recasting layers created by wire electrical discharge machining (WEDM) [5,6]. In addition, a mixed gel is a flexible gel abrasive that can change its shape to fit a working surface during machining; therefore, complex surfaces and irregular holes can be polished efficiency and the method is widely used in industries, such as

automobile, aerospace, bioengineering, mold, and semiconductor [7,8]. However, since an abrasive medium can only create one directional motion using traditional AFM, it is difficulty to achieve uniform roughness in the inner holes or in the complex surfaces after the polishing process using this method [7]. Wang [9,10] developed a core with helical grooves and inserted the core into the hole before machining; an abrasive medium generated a spiral motion on the hole's surface when gel abrasive was passed through the hole during AFM(HP-AFM). Different tangential forces could be generated by an abrasive medium with helical motion, and therefore the finished inner hole was smoother than by using traditional AFM. Due to the multi-directional motion of abrasives, a uniform surface could be easily obtained. Jain et al. [11–14] combined an external rotational mechanism with traditional AFM, the device consisted of a variable frequency drive (VFD) with a motor, and a gear and belt transmission system to setup a rotational abrasive flow finishing (R-AFF) process. This method rotated the workpiece in the AFM process and the abrasive particles abraded the working surface by the helical motion, and therefore, improved the finishing efficiency and material removal rate. The results showed that R-AFF enhanced the roughness improvement rate to 44% and increased the material removal rate up to 81.8% as compared with traditional AFM. In addition, a new rotational abrasive finishing method (R-AF) was setup by Azami et al. [15]. In this nano-finishing method, the stirrer and the workpiece rotated in opposite directions at the same time during the polishing process and gel abrasive was pushed by the stirrer to abrade the working surface; the interior surfaces of the holes reached ultra-low roughness after R-AF. Moreover, magneto-rheological abrasive flow finishing (MR-AFF) has also been shown to achieve better performance of radial force as compared with traditional AFM [16–18]. In MR-AFF, magnetic particles and an electro-magnetic mechanism are applied to increase the abrasive cutting forces in the finishing process, furthermore, steel grits and abrasives are also constrained by the gel, inducing a flexible self-sharpening effect in the finishing process, and therefore, the inner surface can be polished to a very precision level.

Although inserting a helical core in an inner hole generates internal resistance of the gel abrasive passing through the working gap, disadvantages exist including increased working time and decreased polishing efficiency. Moreover, although R-AFF has shown good performance in terms of polishing, the structure of the external rotating device is too complicated and cannot control the rotation direction at the same time, therefore, possibly increasing the cost of the device or limiting the finishing efficiency. In addition, MR-AFF is appropriate for magnetic materials and the simple geometries of the elements. In general, in the abovementioned experiments, it would be necessary to setup a complex fixture to constrain the motion of the gel abrasive using non-magnetic material as the workpieces, or the gel abrasive would easily flow away from the working area because of the small magnetic forces producing in the gap. Therefore, in this study, we develop a servo rotational device with a gear transmission control system to rotate the workpiece in AFM (RDSM-AFM) that allows the gel abrasive to produce different tangential forces on the machining surface, and therefore, the inner holes can be easily polished to good precision by increasing the tangential abrasive forces.

## 2. Materials and Methods

### 2.1. RDSM-AFM Setup and Fixture Design

Traditional AFM includes a hydraulic control system to fix the workpieces and control the reciprocating motion of the abrasive medium in machining and a control box to setup the machining cycles and to define the strokes of hydraulic cylinders before finishing. In addition, since the hydraulic system generates a high temperature when it is actuated, the cooling system is mainly for cooling the motor and hydraulic oil to avoid overheating of the hydraulic system. In this work, we developed a novel mechanism with a gear assembly and a control system to perform spiral flowing paths of abrasive gels by adding a rotating device with a servo motor in AFM. The control system included a PC with an interface control card and software was applied to capture and control the signals of the DC servo

motor. Then, the servo motor drove the transmission gear and passive gear to rotate the workpiece, and two hydraulic cylinders pushed the gel abrasive to produce reciprocating motion in machining; therefore, the inner surfaces of the workpiece could be polished efficiency by the spiral motion of the gel abrasive. A diagram of the rotational mechanism with a DC servo motor and control box in AFM is shown in Figure 1.

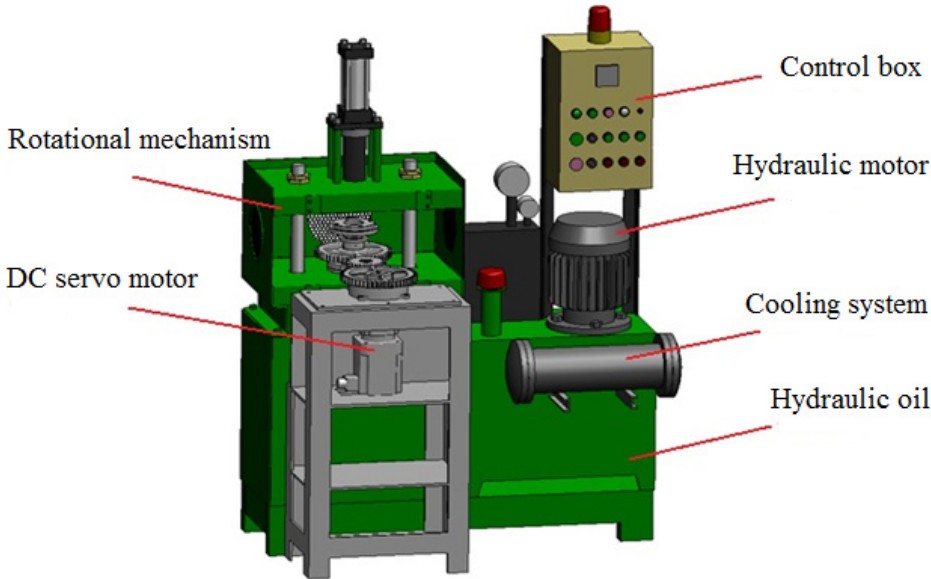

**Figure 1.** AFM with a rotational mechanism and control system.

In this study, the designed fixture was drawn with Solidworks software, as shown in Figure 2, and included an upper and lower fixed block, an upper and lower tool, an upper and lower bearing, a transmission gear, and a passive gear. The parts of the upper tool and the upper fixed block were locked on the upper barrel, while the lower tool and the lower fixed block were locked together on the lower barrel. The transmitted gear was driven by the servo DC motor, and the gear tooth ratio of the transmitted gear and passive gear was 1:5.

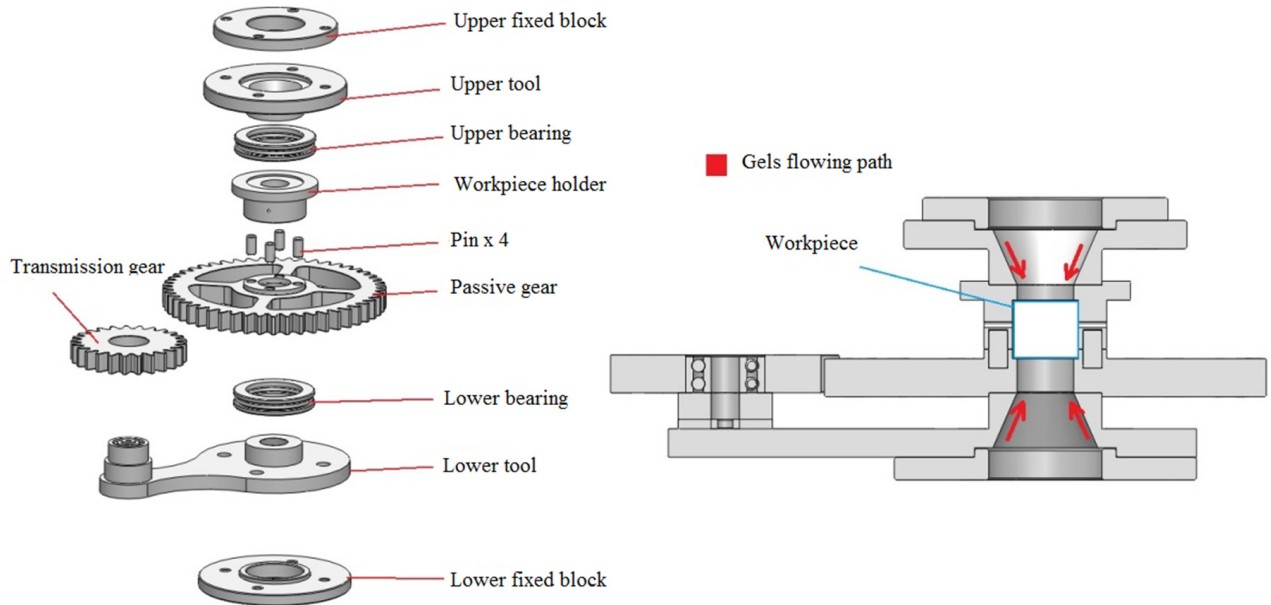

**Figure 2.** Design diagrams of a rotational mechanism with a gear assembly.

In traditional AFM, extruding pressure is applied from the hydraulic system to make the deformable gel abrasive finishing the working surfaces. The medium reciprocating motion exerts axial force ($F_a$) on the workpiece along the axial direction. It also exerts radial force ($F_r$) on the working surface in the finishing process due to the visco-elastic character of the abrasive medium. A diagram of the force components during the traditional AFM is shown in Figure 3. In this study, additional tangential forces ($F_t$) along the tangential direction on the workpiece surface were created by the rotational motion of the gel abrasive, as showed in Figure 4. Therefore, from the finishing forces acting on the working surface, the rotational device in AFM can generate more complex polishing paths than non-rotated AFM. The following mathematical formula describes the total forces ($F_c$) of the abrasive particles in Equation (1):

$$F_c = \sqrt{F_r{}^2 + F_a{}^2 + F_t{}^2} \tag{1}$$

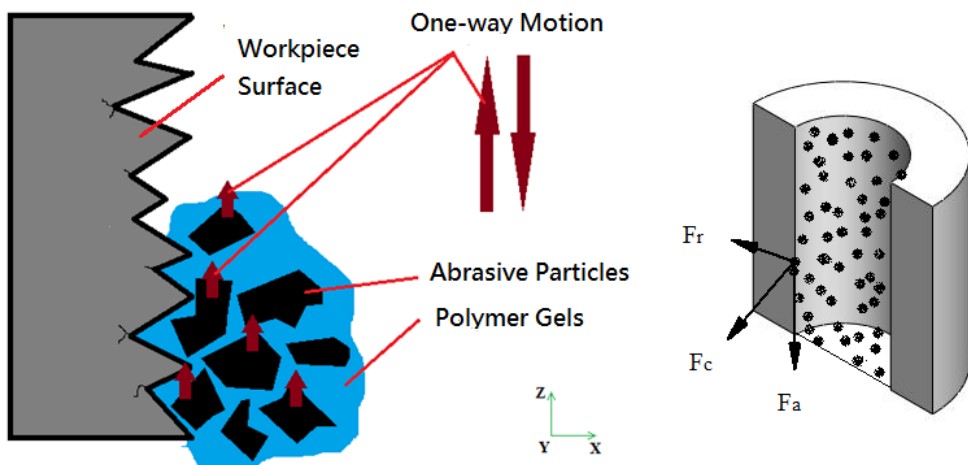

**Figure 3.** Diagram of the force components in traditional AFM.

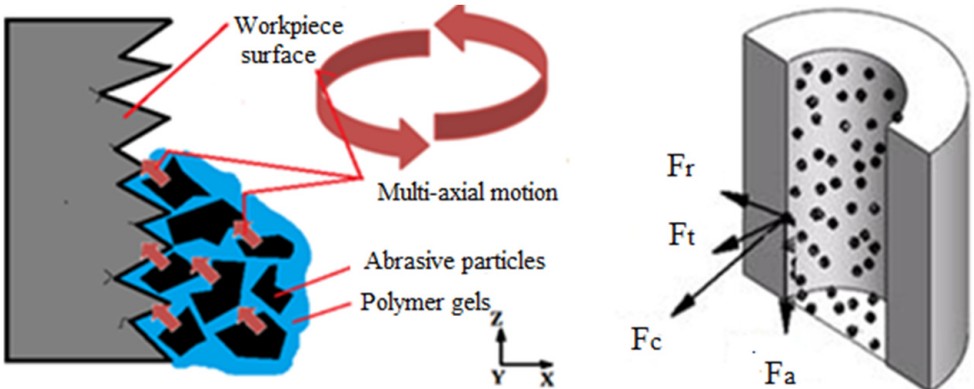

**Figure 4.** Additional tangential forces (Ft) were created by the rotational gel abrasive.

### 2.2. Experimental Materials

In this process, silicone carbon (SiC) was adopted as the abrasive particles and uniformly mixed with silicone gel as the gel abrasive; since silicone gel has good fluidity and can easily change shape to fit a complex surface, the gel abrasive can induce good performances in the surface polishing. Figure 5 illustrates the mixed status of silicone carbon (SiC) and silicone gel. When the hydraulic cylinders were down to the bottom at the beginning stage, the gel medium was filled into the cylindrical barrel first. Then, the designed fixture was installed to prepare the AFM machining process. Table 1 shows the particle size of SiC with the different mesh numbers used in this experiment. In this study, we focused on increasing the roughness improvement rate (RIR) of the inner surfaces in

the radial distribution of the circular holes. Generally, the experiments picked up eight positions in an axial surface to test the surface roughness after RDSM-AFM. Batch circular samples of SKD-11 steel were cut out using the WEDM process. The initially average surface roughness of all the workpieces was approximately 0.60 μm Ra after machining. The outer diameter of each workpiece was 32.0 mm, the inner diameter was 20.0 mm, and the length was approximately 30.0 mm. Figure 6 shows the cutting part of a circular-hole specimen and the design size of the parameters.

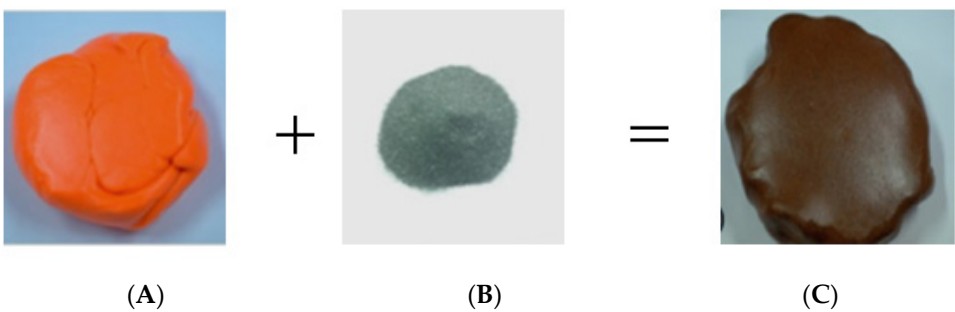

**(A)**  **(B)**  **(C)**

**Figure 5.** Diagrams of making the gel abrasive before RDSM-AFM: (**A**) Silicone gel; (**B**) silicone carbon; (**C**) gel abrasive.

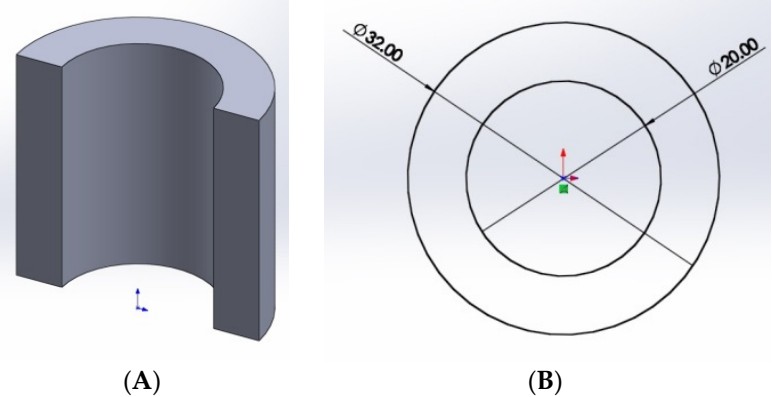

**(A)**  **(B)**

**Figure 6.** Diagrams of the working part by WEDM: (**A**) A cross-section specimen; (**B**) dimensions of the workpiece.

**Table 1.** The mesh number vs. size of abrasive particles of the silicone carbon.

| Mesh Number | Size (μm) |
| --- | --- |
| #60 | 250–300 |
| #100 | 125–150 |
| #220 | 53–75 |
| #500 | 31–37 |

## 3. Results and Discussion

In general, single-factor experiments and optimal design method are often used for experimental verification. Chairman et al. [19,20] used single-factor experiments and Taguchi's method to optimize the process parameters for effective machining in mechanical and abrasive wear performance of composite materials. In this study, single-factor experiments of RDSM-AFM were adopted to investigate the finishing precision during the polishing process. The experimental parameters included rotational speed, rotation direction, abrasive mesh size, silicone gel/abrasive concentration ratio, and machining reciprocating cycle. Table 2 lists the setting levels of the controllable factors. In this study, roughness improvement rate (RIR) values for different parameters were used to evaluate

the polishing effects in RDSM-AFM. To achieve the stated objectives, improvement of the surface roughness was used to demonstrate a good RIR on the inner hole surfaces after polishing. A series of experiments picked up fifteen positions in a radial surface to evaluate the surface roughness. RIR is defined as the following Equation (2):

$$\text{RIR} = \frac{\text{SR}_{\text{origin}} - \text{SR}_{\text{polishing}}}{\text{SR}_{\text{origin}}} \qquad (2)$$

where $\text{SR}_{\text{origin}}$ represents the original surface roughness before RDSM-AFM and $\text{SR}_{\text{polishing}}$ describes the surface roughness after RDSM-AFM polishing. The experimental results of surface polishing with different design parameters are described in the following subsections.

**Table 2.** The design parameters and the setting levels of the controllable factors.

| Factor | Level 1 | Level 2 | Level 3 | Level 4 |
|---|---|---|---|---|
| Rotation direction | 5 (+) | 5 (±) | 10 (+) | 10 (±) |
| Rotational speed (rpm) | 0 | 5 | 10 | 15 |
| Abrasive mesh size | #60 | #100 | #220 | #500 |
| Concentration ratio (wt.%) | 1:1 | 1:1.5 | 1:2 | / |
| Machining reciprocating cycle | 5 | 10 | 15 | 20 |

### 3.1. Effects of the Rotational Speed on Surface Roughness

In order to understand the effect of the rotational speed parameter on finishing, 0 rpm, 5 rpm, 10 rpm, and 15 rpm were applied as the working levels to evaluate the surface roughness of circular holes. The other design parameters were mesh 100# SiC, clockwise rotation of the workpiece, and 1:1 silicone gel/abrasive concentration ratio. Figure 7 presents the polishing effects of the rotational speeds on surface roughness. The results revealed that an increase in rotational speeds resulted in a decrease in surface roughness. The RIR at the rotational speed of 15 rpm was 63%, but the RIR was only 43% at the stationary state within the same machining cycles. Furthermore, the results also showed that surface roughness quickly decreased when only five working cycles were utilized to finish the circular holes. Therefore, high rotational speed resulted in significant tangential forces, creating high polishing efficiency and good surface roughness. Based on the experiments, the machining efficiency and surface roughness at 10 rpm were close to the polishing results at 15 rpm, therefore, high rotational speed achieved high precision of surface roughness by RDSM-AFM, however, the finishing result quickly reached a critical value at some rotational speeds.

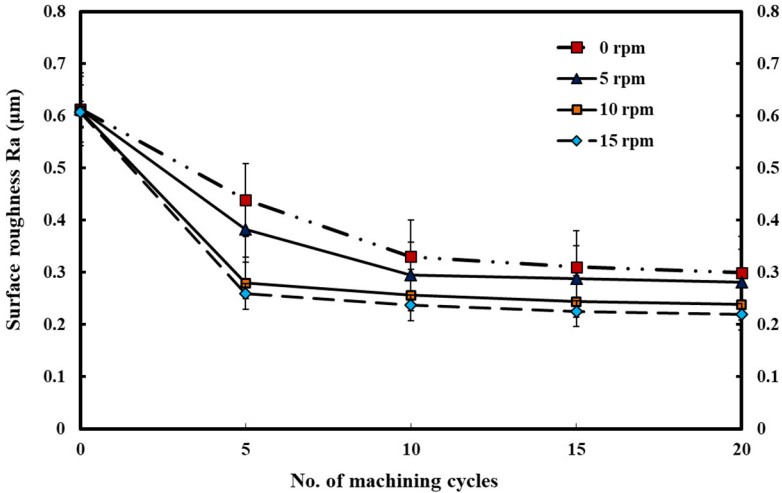

**Figure 7.** Effects of rotational speed on surface roughness measurements.

*3.2. Effects of Rotation Direction on Surface Roughness*

Since different rotation directions can create different finishing paths, surface roughness of the circular holes can possibly be improved when staggered rotation directions are applied in RDSM-AFM. In this study, there were two types of experiments to evaluate the effects of different rotation directions: in one experiment, the workpiece rotation was set only in the clockwise direction, in the other experiment the workpiece rotation was set in clockwise and counterclockwise staggered rotation directions. The rotation changed per 10 reciprocating cycles in machining, in which a positive symbol (+) was defined as clockwise rotation, and a negative symbol (−) was defined as counterclockwise rotation, as shown in Figure 8. Rotational speeds of 5 rpm and 10 rpm were both adopted to evaluate the polishing effects of the workpiece's rotation directions. The results showed that at 5 rpm or 10 rpm the surface roughness quickly decreased to saturated levels in 10 working cycles; the workpiece with a rotational speed of 10 rpm performed better surface roughness than the workpiece with a rotational speed of 5 rpm. However, the workpiece with the staggered rotation direction (±) and the workpiece with one rotation direction (+) almost had the same polishing effects after RDSM-AFM; the results showed that there was no relationship between polishing efficiency of the surface roughness and rotation direction. This is because the rotating direction of the workpiece changed after 10 machining cycles when the staggered rotation direction was set as the parameter, but surface roughness of the circular hole was already at the saturated level after 10 working cycles in the (+) rotating direction, therefore, the workpiece with the staggered rotation direction could not effectively enhance the surface roughness of the circular holes.

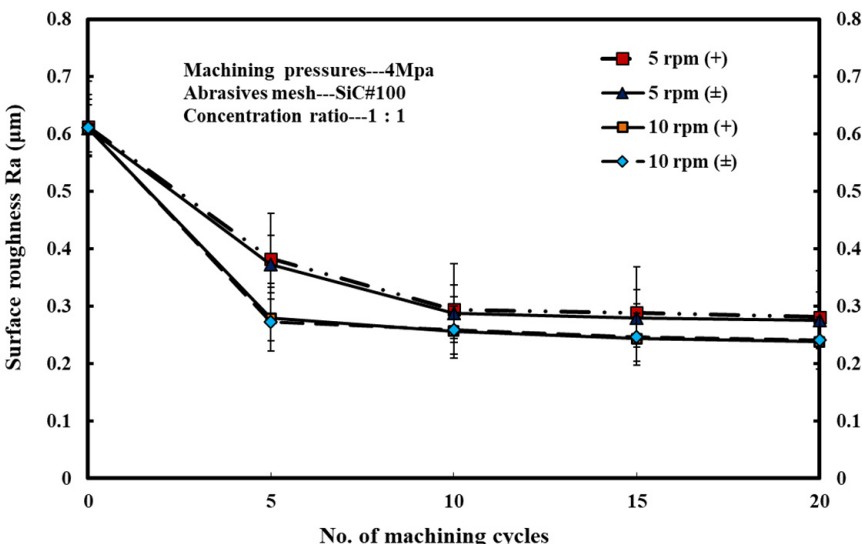

**Figure 8.** Effects of rotation direction on surface roughness.

*3.3. Effects of the Reciprocating Cycles on Surface Roughness*

Reciprocating cycles (or working cycles) in AFM can actually affect polishing efficiency and machining precision [3], therefore, in this study, reciprocating cycles with different rotational speeds in RDSM-AFM were studied to understand the role of the working cycle in machining. Figure 9 displays the polishing effects of the reciprocating cycles with different rotational speeds on surface roughness. According to the results, an increase in the reciprocating cycles in RDSM-AFM resulted in a decrease in surface roughness, however, only five working cycles could generate an obvious change in the surface roughness, but variations in the surface roughness were almost the same from 10 to 20 reciprocating cycles at the same rotational speed. Moreover, surface roughness also decreased with increasing rotational speeds. Except for the surface roughness in 5 machining cycles which decreased significantly during the rotational speeds, surface roughness in the other

working cycles declined slowly when the rotational speeds were increased. Therefore, both results indicated that RDSM-AFM could create good finishing efficiency during a short working time.

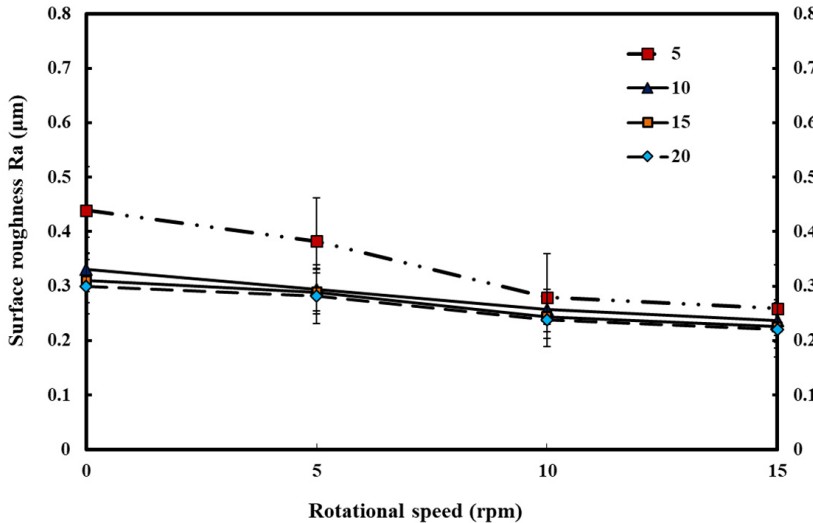

**Figure 9.** Effects of the reciprocating cycles on surface roughness.

### 3.4. Effects of Abrasive Mesh Size on Surface Roughness

Since circular holes were created by WEDM, the surface was full of craters and recasting layers. Usually these defects are not easy to remove by using flexible abrasives, therefore, we assessed the parameter abrasive mesh size to look for good polishing effects during RDSM-AFM. There were four types of abrasive mesh size (#60, #100, #220, #500) adopted in the experiments, and the range of the particle sizes are listed in the Table 1. The effects of the abrasive mesh sizes on surface roughness are shown in Figure 10. The results showed that no matter what type of abrasive mesh was used, the surface roughnesses of the circular holes were reduced when the machining cycles were increased. Furthermore, the small abrasive mesh with large particle size created a better surface roughness than the large abrasive mesh with small particle size. In particular, the RIR of the abrasive mesh #60 reached 61% at the beginning of five reciprocating cycles, but the RIR of the abrasive mesh #500 was only 16% at the same working cycles. Since the recasting layers creating by WEDM are hard with an uneven surface, large abrasive particles can produce high removing capability of the recast layers, therefore, the abrasive mesh #60 could quickly reduce the surface roughness of the circular hole in short time periods.

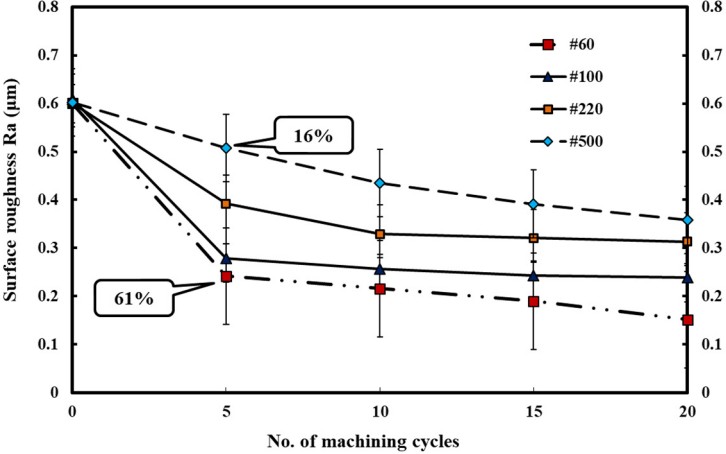

**Figure 10.** Effects of abrasive mesh size on surface roughness.

### 3.5. Effects of the Silicone Gel/Abrasive Concentration Ratio on Surface Roughness

The silicone gel/abrasive concentration ratio is also an important factor to determine polishing precision and machining efficiency. Therefore, three concentration ratios of silicone gel/abrasive, i.e., 1.0:1.0, 1.0:1.5, and 1.0:2.0 were taken for this parameter to estimate the machining effects of RDSM-AFM. Figure 11 presents the polishing effects of the silicone gel/abrasive concentration ratio on surface roughness. The results illustrate that an increase of the abrasive concentrations resulted in an obvious decrease in the surface roughness; the RIR of the concentration ratio 1.0:1.0 reached 75% during 20 reciprocating cycles, however, the RIR of the concentration ratios 1.0:1.5 and 1.0:2.0 reached 82% after 20 machining cycles. The reason was that the polishing forces acting on the surface were proportional to the abrasive concentration of the medium; a high concentration of the abrasive could perform high polishing effects. Sharp areas on the workpiece surface could be quickly removed by using a high concentration abrasive at the beginning stage, however, after that, the gel abrasive could not create the efficient polishing effect when the polished surface became a little flat, and therefore the surface roughness of circular holes quickly went to the saturated state at five machining cycles. In addition, based on the experimental results, the machining precision at a silicone gel/abrasive concentration ratio of 1.0:1.5 was very close to the silicone gel/abrasive concentration ratio of 1.0:2.0; therefore, a silicone gel/abrasive concentration ratio of 1.0:1.5 was chosen as the working parameter to excute the following experiments.

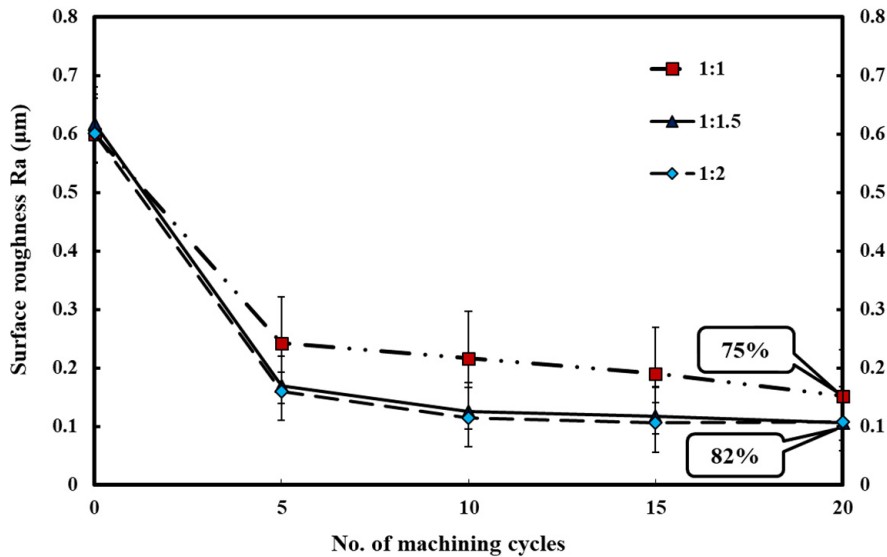

**Figure 11.** Effects of concentration ratios of silicone gel and abrasive on surface roughness.

### 3.6. Effect of AFM Methods on Surface Roughness

In order to verify that RDSM-AFM had good polishing effects as compared with other AFM methods, traditional AFM and HP-AFM (a helical core was inserted into the hole) were utilized as the machining methods to evaluate the polishing results on surface roughness. Abrasive mesh #60 and a silicone gel/abrasive concentration ratio of 1:1.5 were selected as the fixed parameters in this experiment, and a rotational speed of 15 rpm was also applied as the machining parameter in RDSM-AFM. Figure 12 displays the polishing effects of the AFM methods on surface roughness after 20 working cycles. The results showed that surface roughness of the circular holes were reduced by increasing the reciprocating cycles for all AFM methods, however, RDSM-AFM could quickly reduce the surface roughness of the circular hole during the machining and the RIR increased to 80% at five working cycles, however, the RIR of traditional AFM and HP-AFM only reached 25% and 38% at five machining cycles. The results also revealed that the RIR of traditional AFM only reached 58% at 20 reciprocating cycles, but the RIR of HP-AFM and RDSM-AFM reached high

finishing effects of 84% and 87% at the same working cycles. Therefore, the results proved that the rotating motion of the gel abrasive in AFM could create the additional tangential forces and achieve excellent polishing effects, however, RDSM-AFM performed better than HP-AFM.

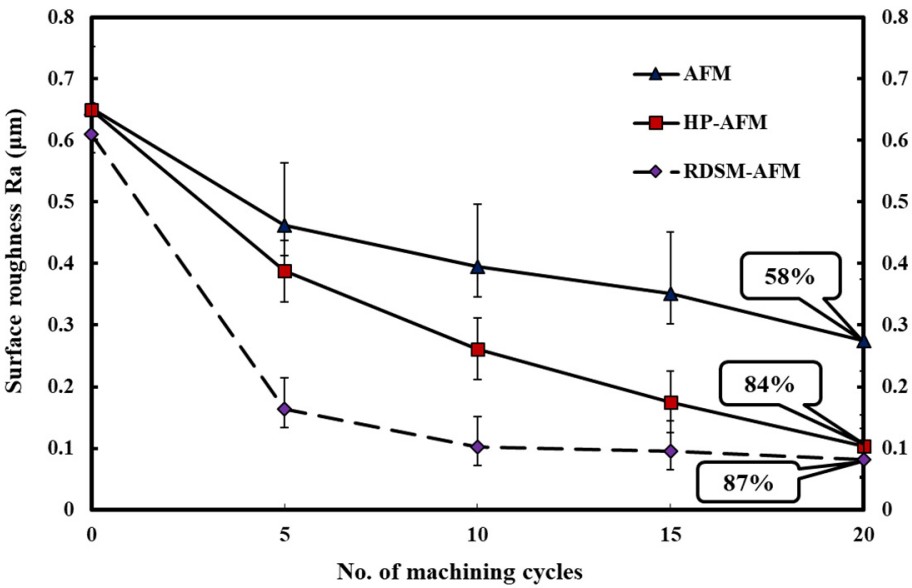

**Figure 12.** Effects of AFM methods on surface roughness.

*3.7. Finishing Results of the Polished Surface*

The function of RDSM-AFM was demonstrated in the above section. Further, the surface characteristics of workpieces were illustrated herein to compare the the same position before and after polishing. Figure 13 shows the photos taken by the scanning electron microscope (SEM), Figure 13a presents the SEM micrography of an initial workpiece surface after WEDM, and Figure 13b displays the SEM micrography of the workpiece surface after polishing byRDSM-AFM. The surface cut by WEDM was full of craters and recasting layers, as shown in Figure 13a, however, these defects were removed by the RDSM-AFM polishing process. Therefore, combining the rotating device with AFM indeed enhanced the polishing effect efficiency.

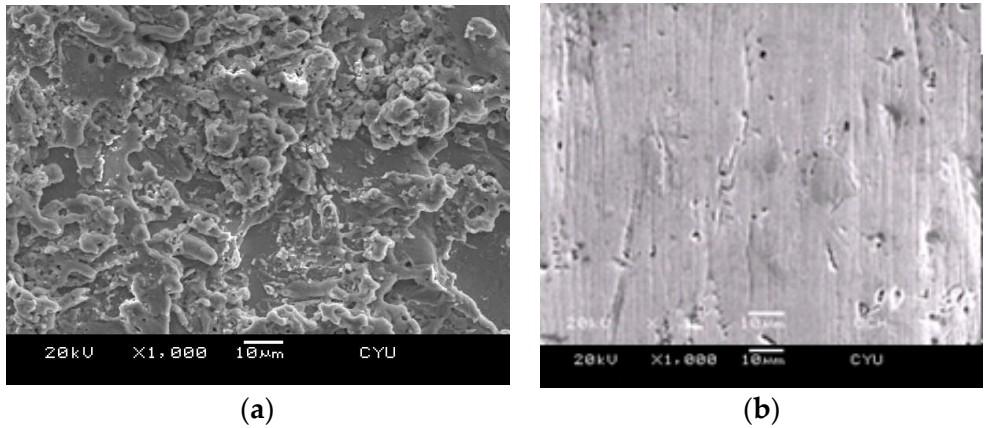

| (**a**) | (**b**) |

**Figure 13.** Thecharacteristics of SEM micrography (×1000) of the workpiece surface: (**a**) SEM photo during WEDM; (**b**) SEM photo after polishing by RDSM-AFM.

## 4. Conclusions

In this work, a novel mechanism with a gear assembly and a control system was developed to perform the helical flowing paths of gel abrasive by adding a rotating device with a servo motor in traditional AFM. The main conclusions are summarized as follows.

1.  Surface roughness of the circular holes was decreased by increasing the rotational speeds in RDSM-AFM, however, the finishing results quickly reached a critical value at rotational speeds of 10 rpm and 15 rpm.
2.  Since the workpiece changed rotating direction after 10 machining cycles, and surface roughness of the circular hole was already at a saturated level, therefore, the workpiece with staggered rotation directions could not effectively enhance surface roughness.
3.  Surface roughness of the inner hole could quickly decrease to the saturated values only after 5 machining cycles, it indicated that RDSM-AFM could effectively reduce the cost of the surface polishing.
4.  The small abrasive mesh #60 produced better surface roughness than the large abrasive mesh #500, the RIR of the abrasive mesh #60 reached 61% at five reciprocating cycles, but the RIR of the abrasive mesh #500 only reached 16% at the same working cycles.
5.  High silicone gel/abrasive concentration ratios resulted in good polishing effect in RDSM-AFM, the RIR of the concentration ratios 1.0:1.5 and 1.0:2.0 reached 82% at the rotational speed of 10 rpm.
6.  RDSM-AFM performed significantly better than traditional AFM and HP-AFM in short time periods; the RIR of RDSM-AFM almost reached 80% at five working cycles with a rotational speed of 15 rpm, but the RIR of traditional AFM and HP-AFM only reached 25% and 38% under the same conditions.

**Author Contributions:** Conceptualization, K.-C.C. and A.-C.W.; methodology, A.-C.W. and C.-Y.H.; validation, K.-Y.C., K.-C.C. and C.-Y.H.; investigation, K.-C.C.; resources, A.-C.W.; data curation, K.-Y.C. and K.-C.C.; writing—original draft preparation, K.-C.C. and A.-C.W. All authors have read and agreed to the published version of the manuscript.

**Funding:** This research was supported by the Ministry of Science and Technology, Taiwan, the funding no. was MOST 105-2221-E-231-003–MY2.

**Institutional Review Board Statement:** Not applicable.

**Informed Consent Statement:** Not applicable.

**Data Availability Statement:** Not applicable.

**Acknowledgments:** The authors would like to thank the Ministry of Science and Technology, Taiwan for financial support to this study.

**Conflicts of Interest:** The authors declare no conflict of interest.

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
