# Peer review of "Study of the Polishing Characteristics by Abrasive Flow Machining with a Rotating Device"

_processes, doi:10.3390/pr10071362_

Round 1

Reviewer 1 Report

In this article authors have proposed and investigated a new rotational abrasive flow machining method. In my opinion this paper deals with very important and interesting topic related to the development of a novel mechanism in AFM. Before further consideration, some small issues should be considered:

1. The paper contains some small grammatical errors which needs to be thoroughly checked and corrected e.g. lines 115, 139, 236, 284.

2. A more detailed description of the planned experiments should be provided e.g. in the form of a table with a clear division into test types and presentation of a set of parameters (input variables) used in each of them.

3. Were the tests or measurements repeated? There are no error bars on the diagrams.    

4. Why did the authors use the same machining pressure value in each of the experiments? On what basis was this value selected? Can the change of machining pressure have a significant influence on the obtained value of the Ra parameter?

5. Some figures seems not clear, e.g. scale in figure 13 b is not visible.

6. In further research, I would also suggest including a 3D surface analysis using spatial parameters, e.g. Sa.

Author Response

Thanks for the reviewer opinions, all the questions are replied at the following:

  1. The paper contains some small grammatical errors which needs to be thoroughly checked and corrected e.g. lines 115, 139, 236, 284.

Ans. Thanks reviewer for the correcting. All of them has been modified in this manuscript with the new version.

  1. A more detailed description of the planned experiments should be provided e.g. in the form of a table with a clear division into test types and presentation of a set of parameters (input variables) used in each of them.

Ans. The planned experiments has been provided in Table 2 to show the design parameters and setting levels.

  1. Were the tests or measurements repeated? There are no error bars on the diagrams.

Ans.1. All the experiments were repeated for three times. 2. Error bars were already added in the Fig. 7 to Fig. 12.

  1. Why did the authors use the same machining pressure value in each of the experiments? On what basis was this value selected? Can the change of machining pressure have a significant influence on the obtained value of the Ra parameter?

Ans. The direct influence of machining pressure is the faster flowing speed of abrasive medium in AFM process to shorten the machining time. Normally, the maximum machining pressure of AFM device is 6 MPa in our lab. Due to the potential risk of medium leakage, therefore, considering to adopt 4 MPa pressure for AFM machining is stable and safety in our experience.

  1. Some figures seems not clear, e.g. scale in figure 13 b is not visible.

Ans. The figure 13 b has been modified to show the detail value of scale.

  1. In further research, I would also suggest including a 3D surface analysis using spatial parameters, e.g. Sa.

Ans. Thanks for the good suggestion, we will consider to appoint 3D surface analysis in further research.

Reviewer 2 Report

1. In abstract Conclusion not included.
2. Kindly check once all keywords will be in the running text.
3. In introduction part any human samples have been tested.
4. In material and methods kindly include the preparation procedure.
5. P value is missing.
6. kindly include the following reference in the paper.
Chelliah Anand Chairman, Manickam Ravichandran, Vinayagam Mohanavel,  Ahmad Rashedi, Ibrahim M. Alarifi, Irfan Anjum Badruddin, Ali E. Anqi and Asif Afzal ‘Mechanical and Abrasive Wear Performance of Titanium Di-Oxide Filled Woven Glass Fibre Reinforced Polymer Composites by Using Taguchi and EDAS Approach, Materials, 14(18), 5257, 1-15, 2021.

T. Sathish, ‘Exploration on surface roughness in abrasive water jet cutting of AA6063-TiC Composites for Vehicle Structural Applications’, International Journal of Vehicle structures and systems, Vol. 11, Issue 4, Sep 2019, pp. 417-421, 2019. 

Author Response

Reviewer’s Comments and the responds for the Authors

  1. In abstract Conclusion not included.

Ans. The briefly conclusion and result has been written in abstract.
2. Kindly check once all keywords will be in the running text.

Ans. Thanks reviewer for the correcting. The all keywords has been checked and modified.
3. In introduction part any human samples have been tested.

Ans. The update manuscript was modified in introduction.
4. In material and methods kindly include the preparation procedure.

Ans. The detail illustration of the preparation procedure was adding in section 2.2.
5. P value is missing.

Ans. The direct influence of machining pressure is the faster flowing speed of abrasive medium in AFM process to shorten the machining time. Normally, the maximum machining pressure of AFM device is 6 MPa in our lab. Due to the potential risk of medium leakage, therefore, considering to adopt 4 MPa pressure for AFM machining is stable and safety in our experience.
6. kindly include the following reference in the paper.

Ans. The detail illustration of the reference [19-20] was adding in section 3.
